

# Effect of music interventions on anxiety during labor: a systematic review and meta-analysis of randomized controlled trials

Hsin-Hui Lin[1], Yu-Chen Chang[1], Hsiao-Hui Chou[1], Chih-Po Chang[1], Ming-Yuan Huang[2], Shu-Jung Liu[3], Chin-Han Tsai[4], Wei-Te Lei[5,6] and Tzu-Lin Yeh[7,8]

[1] Department of Family Medicine, MacKay Memorial Hospital, Taipei, Taiwan
[2] Department of Hospice and Palliative Care, MacKay Memorial Hospital, Tamsui Branch, New Taipei City, Taiwan
[3] Department of Medical Library, MacKay Memorial Hospital, Tamsui Branch, New Taipei City, Taiwan
[4] Department of Gynecology and Obstetrics, Hsinchu MacKay Memorial Hospital, Hsinchu, Taiwan
[5] Department of Pediatrics, Hsinchu MacKay Memorial Hospital, Hsinchu, Taiwan
[6] Graduate Institute of Clinical Medical Sciences, College of Medicine, Chang Gung University, Taoyuan City, Taiwan
[7] Department of Family Medicine, Hsinchu MacKay Memorial Hospital, Hsinchu, Taiwan
[8] Institute of Epidemiology and Preventive Medicine, National Taiwan University, Taipei, Taiwan

Corresponding authors
Wei-Te Lei, weite.lei@gmail.com
Tzu-Lin Yeh, 5767@mmh.org.tw

## ABSTRACT

**Background:** Anxiety is commonly experienced during the delivery process and has shown to have adverse effects on maternal and infant health outcomes. Music interventions tend to reduce the effects of anxiety in diverse populations, are low cost, are easily accessible, and have high acceptability. The aim of this review and meta-analysis was to assess the effectiveness of music interventions in reducing anxiety levels among women during labor.

**Methods:** Seven databases from inception to the end of December, 2018, without any language or time restriction including Embase, PubMed, the Cochrane Library, the Cumulative Index to Nursing and Allied Health, PsycINFO, Airiti Library, and PerioPath: Index to Taiwan Periodical Literature were searched using key terms related to pregnancy, anxiety, and music. Randomized controlled trials that assessed the effect of music during labor and measured anxiety levels as an outcome were included. Meta-analyses were conducted to assess anxiety reduction following a music intervention compared to that after placebo treatment.

**Results:** A total of 14 studies that investigated a total of 1,310 participants were included in this review. The meta-analyses indicated that those in the intervention group had a significant decrease in anxiety scores (standardized mean difference = −2.40, 95% confidence interval (CI) [−3.29 to −1.52], $p < 0.001$; $I^2 = 97.66\%$), heart rate (HR) (difference in means = −3.04 beats/min, 95% CI [−4.79 to −1.29] beats/min, $p = 0.001$; $I^2 = 0.00\%$), systolic blood pressure (SBP) (difference in means = −3.71 mmHg, 95% CI [−7.07 to −0.35] mmHg, $p = 0.031$; $I^2 = 58.47\%$), and diastolic blood pressure (DBP) (difference in means = −3.54 mmHg, 95% CI [−5.27 to −1.81] mmHg, $p < 0.001$; $I^2 = 0.00\%$) as compared to the women in the control group.

**Conclusions:** Music interventions may decrease anxiety scores and physiological indexes related to anxiety (HR, SBP, and DBP). Music interventions may be a good non-pharmacological approach for decreasing anxiety levels during labor.

# INTRODUCTION

During pregnancy and labor, women experience psychological and physiological changes that generate stress (*Cardwell, 2013*). With the progression of labor, women experience increasing anxiety during labor; which has a negative effect on the mother as well as the newborn baby (*Zijlmans et al., 2017*). The prevalence of anxiety among prenatal women is higher than in the general population (27% compared to 5%), and there were significantly higher complication rates in anxious women (*Zhao & Zhu, 1999*). There might be more postpartum psychiatric symptoms, decreased sexual functioning, less willing for a next child, and poor mother–infant connection when mothers had more negative childbirth experiences during labor (*Goodman, 2004*). In addition, high anxiety levels may also lead to negative outcomes in women undergoing cesarean section (CS), including higher analgesic consumption, elevated blood pressure (BP), elevated heart rate (HR), increased cortisol level, reduced immune response, slower wound healing, and higher infection risk (*Gorkem et al., 2016*; *Hepp et al., 2016*; *Scott, 2004*).

Music is an ancient healing practice that can inspire the soul as well as improve immunity, forming a powerful therapy (*Lane, 1992*). Listening to music reduces the catecholamine levels, thus improving physical health status, decreasing stress hormones, and stabilizing vital signs (*Liu, Chang & Chen, 2010*; *Mok & Wong, 2003*). Moreover, music interventions have an effect in decreasing pain, anxiety, and analgesic consumption in previous studies (*Ikonomidou, Rehnström & Naesh, 2004*; *Siedliecki & Good, 2006*; *Smolen, Topp & Singer, 2002*). In medical care, music interventions may include music listening initiated by patients, music medicine (listening to prerecorded music offered by medical staff for symptom management), and music therapy (individualized music interventions including listening to live, or prerecorded music, playing instruments and composing music offered by a trained therapist) (*Bradt et al., 2015*).

Most relaxants and antidepressants cross the placental barrier and have negative effects on the fetus; therefore, establishing alternative non-pharmaceutical interventions to reduce anxiety in pregnant women is important. A recent Cochrane Database Systematic Review has shown that music-based interventions may reduce anxiety during pregnancy (*Corbijn Van Willenswaard et al., 2017*). However, the evidence regarding the efficacy of music interventions during labor on the reduction of anxiety is inconclusive. One study has shown that music intervention has a significant positive effect on anxiety and pain during the latent phase of labor (*Liu, Chang & Chen, 2010*). Another study has reported that music intervention during CS does not significantly change the anxiety score (*Reza et al., 2007*). One publication has revealed significantly lower anxiety and higher

satisfaction level after music intervention during CS; however, there was no significant difference in any of the physiological indexes (*Chang & Chen, 2005*). Therefore, we conducted this systematic review and meta-analysis to evaluate the effectiveness of music interventions in reducing the anxiety levels of women undergoing labor.

# MATERIALS AND METHODS

## Search strategy

The review protocol has been registered in the PROSPERO International Prospective Register of Systematic Reviews (registration number: CRD42018108267) and was written according to the preferred reporting items for systematic reviews and meta-analyses statement (*Liberati et al., 2009*) (Table S1).

Seven databases were searched from inception to the end of December 2018, without any language or time restriction including Embase, PubMed, the Cochrane Library, the cumulative index to nursing and allied health, PsycINFO, Airiti Library, and PerioPath: Index to Taiwan Periodical Literature. A professional librarian reviewed the terms and organized them in an optimal manner to make the search strategy sensitive and specific.

The search key terms were "labor," "music," "anxiety," and "stress." Keywords were combined using Boolean searches, and the search was performed using keywords, Boolean operators, and MeSH descriptors. The details of the search strategy have been described in Table S2.

## Selection of studies

Two authors (HHL and MYH) screened the title and abstract of each study that met the inclusion criteria independently, and the controversies were resolved through discussions with the third author (TLY). Two independent reviewers (HHL and TLY) assessed the eligibility of each publication after the initial search. The inclusion criteria of selected randomized control trials (RCT) were as follows: (1) studies on women who underwent vaginal or CS delivery, either nulliparous or multiparous, of normal term, singleton gestation; (2) at least one treatment group wherein music intervention was applied during the labor process; (3) inclusion of a placebo group as control; and (4) reporting of anxiety status after the intervention. We excluded the following: (1) studies on women with deafness (unless corrected with a hearing aid), high risk of pregnancy, or severe psychiatric disorder; (2) duplicate publications; (3) crossover study designed trials; and (4) studies with an effective intervention as control arm rather than a placebo.

## Data extraction

Two authors (HHL and WTL) independently extracted the data, and the inconsistencies were resolved through discussion. The following information was collected (Table 1): first author's name, year of publication, country of publication, number of participants, age of participants, number of participants in the intervention and control groups, details of the intervention (including the music types, time, and duration), and clinical outcome measures (including the time of the outcome in relation to the treatment). The primary outcome was the anxiety status, measured using the recognized rating scales, including

**Table 1** Characteristics of randomized controlled trials investigating the effect of music intervention on anxiety during labor.

| Study | Country | Participants | Age (Mean ± SD) (I vs. C) | I:C | Intervention | Outcome measures | Findings |
|---|---|---|---|---|---|---|---|
| Choubsaz et al. (2018) | Iran | 60 low-risk pregnant women, ASA class I and II undergoing elective CS | 27.1 ± 4.94 vs. 26.6 ± 5.59 | 30:30 | The "Motivation" piece, a sedative musical piece, was played through a headphone during the surgery | STAI before and after OP | Significant differences between STAI in music (21.83 ± 11.9 vs. 13 ± 8.02) and control groups (24.4 ± 11.89 vs. 16.6 ± 8.14) pre and post-test ($p = 0.001$) |
| Hepp et al. (2018)* | Germany | 304 low-risk pregnant women undergoing primary CS | 33.5 ± 5.4 vs. 33.7 ± 5.4 | 154:150 | Slow tempo music from one (15 tracks) of four self-selected genres via loudspeakers started when entering the OR | VAS-A, STAI, salivary cortisol and salivary alpha amylase at admission, skin suture, and 2 h after OP; HR, SBP and DBP at skin incision and 2 h after OP | At skin suture, significantly lower STAI ($p = 0.004$) and VAS-A ($p = 0.018$). 2 h after OP, lower VAS-A ($p = 0.018$); salivary cortisol increased from admission to skin suture ($p = 0.043$); lower SBP ($p = 0.002$) and HR ($p = 0.049$) at skin incision |
| Gokyildiz Surucu et al. (2018)* | Turkey | 50 low-risk primigravid women | NR | 25:25 | Music was played in Acemasiran mode with earplugs for 3 h (20 min of listening with 10-min breaks) during the active phase | STAI, faces anxiety scale 30 min after intervention, RR, HR, SBP, DBP, dilatation, effacement, fetal HR, period of contraction, frequency of contraction at the end of 1st, 3rd, 5th, 7th h of labor | The women who listened to music during labor had lower anxiety levels, evaluated the labor as easier, had longer periods of contraction, and their labor progressed faster |
| Wan & Wen (2018)* | China | 119 low-risk primigravid women, singleton, and expected to have NSD | 25.57 ± 3.11 vs. 26.02 ± 2.90 | 60:59 | Relaxing, soft and regular rhythmic music recommended to participants was played with a 20-min break for every 2 h during the active phase | VAS-A (1, 4, 8, 16, 24 h after intervention) | VAS-A scores significantly different from those in the controls at all-time points (all $p < 0.05$) |
| Karkal, Kharde & Dhumale (2017)* | India | 60 primigravid women who were in the active phase of labor | NR | 30:30 | Music was administered in the first stage of labor | Zung's SAS (pre-test and post-test) | The mean post-test score of anxiety between the experimental and control groups was 40.01 ($p < 0.001$) |

| Study | Country | Participants | Age (Mean ± SD) (I vs. C) | I:C | Intervention | Outcome measures | Findings |
|---|---|---|---|---|---|---|---|
| Simavli et al. (2014a)* | Turkey | 132 low-risk primigravid women, singleton, expected to have NSD | 25.06 ± 4.33 vs. 25.09 ± 4.53 | 67:65 | Self-selected music (relaxing, regular rhythmic patterns) were played all the time with 20-min break for every h via headphones since two cm cervical dilatation to first 2 h of the active phase | VAS-A, SBP, DBP, HR (before music, latent phase, active phase, second stage and 2 h postpartum); analgesic requirement | A significantly lower level of anxiety ($p < 0.001$), maternal hemodynamics and fetal HR ($p < 0.01$) in the interventional group at all stages of labor and analgesic requirement postpartum ($p < 0.001$) |
| Simavli et al. (2014b)* | Turkey | 141 low-risk primigravid women, singleton, expected to have NSD | 24.17 ± 3.22 vs. 23.39 ± 3.88 | 71:70 | Self-selected music (relaxing, regular rhythmic) played all the time with a 20-min break for every 2 h since two cm cervical dilatation to the end of the third stage | VAS-A (1, 4, 8, 16, 24 h postpartum), VAS-S (2, 12, 24 h postpartum) | Significantly lower postpartum anxiety at all-time intervals (1, 4, 8, 16, and 24 h, $p = 0.001$). Significantly lower satisfaction rate ($p = 0.001$) |
| Kushnir et al. (2012) | Israel | 60 low-risk pregnant women, undergoing an elective CS for medical reasons only | 32.0 ± 3.97 vs. 32.13 ± 4.79 | 28:32 | Patient-preferred music (light classical music or Israeli tunes) were played for 40 min, using a Discman with earphones, while lying on their beds before OP | 1. Mood states scale 2. Perceived threat of surgery scale 3. SBP, DBP, HR, RR (before and after 40 min of music listening) | Significant increase in positive emotions and a significant decline in negative emotions and perceived threat of the situation. Significant reduction in SBP, increase in DBP and RR |
| Li & Dong (2012)* | China | 60 low-risk pregnant women, ASA class I and II, undergoing elective CS | NR | 30:30 | Self-chosen Chinese classical music was played for 30 min before OP and was continued after anesthesia | The Zung's SAS, total power, LF, HF, and LF/HF ratio in HRV (at the preoperative visit and just before OP) | The mean HRV was significantly less, the mean HF value was significantly increased, and the mean anxiety score was significantly decreased |
| Blackburn et al. (2011) | United States | 50 low-risk pregnant women, undergoing an elective CS, singleton | NR | 25:25 | Self-selected music provided through MP3 player with programmed genres of music administered for 30 min prior to and after their CS | STAI after intervention | The intervention of patient-selected music before and after CS will reduce the anxiety levels in the patients undergoing CS ($p < 0.05$) |

(Continued)

| Study | Country | Participants | Age (Mean ± SD) (I vs. C) | I:C | Intervention | Outcome measures | Findings |
|---|---|---|---|---|---|---|---|
| *Liu, Chang & Chen (2010)* | Taiwan | 60 low-risk primigravid women, singleton, expected to have NSD | 26.63 ± 4.02 vs. 27.60 ± 4.34 | 30:30 | Self-chosen music (include classical, light, popular, crystal, or Chinese religious music) was played at least 30 min during the latent phase and active phase | VAS-A, finger temperature (before and after 30 min of music listening during the latent and active phases) | The experimental group had significantly lower pain, anxiety and a higher finger temperature during the latent phase |
| *Reza et al. (2007)** | Iran | 100 low-risk pregnant women, ASA class I, undergoing elective CS under general anesthesia | 26 ± 5.19 vs. 25 ± 4.23 | 50:50 | Intra-OP music (soft instrumental, including 15 segments of a Spanish style guitar not selected by the patients) | VAS-A immediately in PACU and at 0.5, 1, 2, 4, and 6 h postoperatively | There were no significant differences in terms of the post-OP anxiety in PACU and at 0.5, 1, 2, and 4 h post-OP |
| *Chang & Chen (2005)** | Taiwan | 64 low-risk pregnant women, undergoing elective CS, singleton, received spinal or epidural anesthesia | 30.31 ± 4.16 vs. 32.31 ± 4.48 | 32:32 | Self-selected music (Western classical, new age, or Chinese religious music) administered for at least 30 min from the start of anesthesia until the end of OP | VAS-A, SpO2, finger temperature, RR, HR, SBP, DBP prior to anesthesia, the end of maternal contact with the neonate, and after completion of the skin suture | Significantly lower anxiety level and a higher level of satisfaction regarding the CS. No significant differences were found between the two groups in any of the physiological indexes |
| *Lee et al. (2004)** | Korea | 50 low-risk pregnant women undergoing an elective CS | 28.1 ± 7.0 vs. 29.7 ± 5.1 | 25:25 | Patients wore the headphones as soon as they entered the OR and listened to the music (the Four Seasons Vivaldi and other five songs (e.g., lullaby–Mozart) three times repeat) | SBP, DBP, HR, intra-OP awareness, post-OP explicit and implicit memory, plasma cortisol (1 min before, 1 min after, 3 min after intubation; 1 min after, 10 min after delivery; 5 min after extubation) | Music significantly decreased SBP and HR at 1 min after intubation and 5 min after extubation, increased hit ratio of the implicit memory test, decreased cortisol at 30 min after intubation and 10 min after arriving in the recovery room |

**Notes:**

ASA, American Society for Anesthesiology; C, control; CS, caesarean section; DBP, diastolic blood pressure; h, hour; HF, low-frequency power; HR, heart rate; HRV, heart rate variability; I, intervention; LF, low-frequency power; min, minute; NR, not reported; NSD, normal spontaneous delivery; OP, operation; OR, operation room; PACU, post-anesthesia care unit; RR, respiratory rate; SAS, self-rating anxiety scale; SBP, systolic blood pressure; SpO2, pulse hemoglobin oxygen saturation; STAI, state-trait anxiety inventory; VAS, visual analog scale; VAS-A, VAS for anxiety; VAS-S, VAS for satisfaction.

* Studies included in meta-analysis.

visual analog scale for anxiety (VAS-A), state-trait anxiety inventory (STAI), and self-rating anxiety scale (SAS). VAS-A is a scale comprising a 10 cm line on which the participant marks her current degree of anxiety, with the left end of the line being labeled "no anxiety" and the right end being labelled "maximum anxiety." The scale is scored by measuring the distance of the mark from the left end (*Hornblow & Kidson, 1976*). The outcome was measured after the delivery process. The secondary outcomes were physical signs, including systolic blood pressure (SBP), diastolic blood pressure (DBP), and HR.

## Assessment of the risk of bias in the included studies

Three reviewers (YCC, HHC, and CPC) independently assessed the risk of bias using the Cochrane Review risk of bias assessment tool. Conflicts were resolved through discussion. We assessed sequence generation (selection bias), allocation sequence concealment (selection bias), blinding of participants and personnel (performance bias), blinding of outcome assessment (detection bias), incomplete outcome data (attrition bias), selective outcome reporting (reporting bias), and other potential sources of bias (Table S3).

## Statistical analyses

Comprehensive meta-analysis software (version 3.0, Biostat, Englewood, NJ, USA) was used for the analyses. There was significant (and expected) heterogeneity among the studies; therefore, a random effects model was selected (*Higgins & Thompson, 2002*). Pooled estimates were calculated with 95% confidence intervals (CIs). Statistical heterogeneity was assessed using $I^2$ and Cochran's Q-tests. A *p*-value of <0.10 for the $\chi^2$ test of the Q statistic or an $I^2$ of >50% was considered indicative of statistically significant heterogeneity (*Higgins et al., 2003*). A sensitivity analysis was performed by repeating the analysis after sequential exclusion of one study at a time to observe the effect on the overall results. Potential publication bias was evaluated by observing the symmetry of the funnel plots and using Egger's test (*Egger et al., 1997*). Furthermore, subgroup analysis was performed to further analyze the effects of clinical variables as the possible origins of heterogeneity, such as ways of delivery, and types of music. Finally, meta-regression analyses were performed only when the data could be assessed throughout more than five trials.

# RESULTS

## Description of studies and quality assessment

Database searches identified 1,460 studies, of which 299 were duplicates (Fig. 1). Finally, 14 publications were included in our qualitative synthesis and critical review (Table 1) (*Blackburn et al., 2011*; *Chang & Chen, 2005*; *Choubsaz et al., 2018*; *Gokyildiz Surucu et al., 2018*; *Hepp et al., 2018*; *Karkal, Kharde & Dhumale, 2017*; *Kushnir et al., 2012*; *Lee et al., 2004*; *Li & Dong, 2012*; *Liu, Chang & Chen, 2010*; *Reza et al., 2007*; *Simavli et al., 2014a*, *2014b*; *Wan & Wen, 2018*). All these studies, except one from the US and one recently published study from Germany, were conducted in Asia; of these, six were performed in Middle Asia. The sample sizes ranged from 50 to 304. The mean age of the participants ranged from 23.8 to 33.6 years. All the studies included women who were at low risk. Eight studies investigated pregnant women undergoing CS.

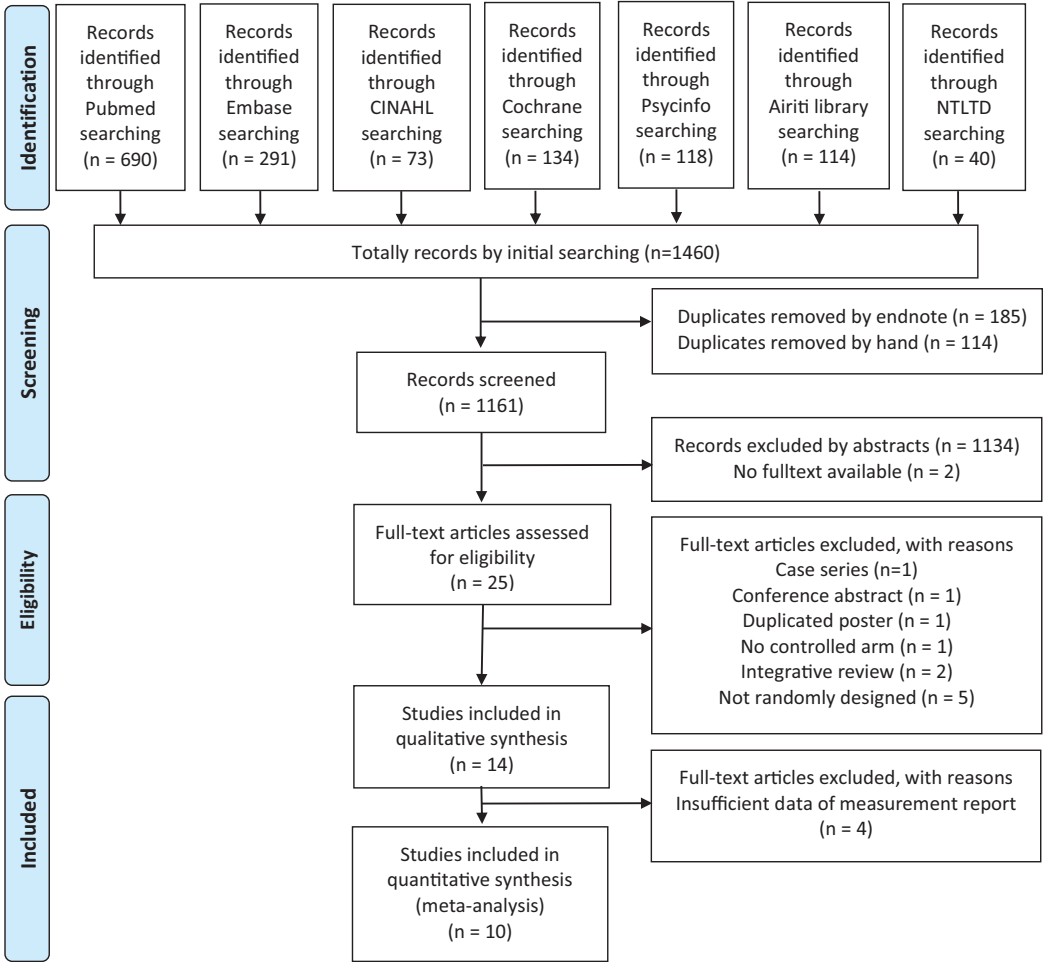

**Figure 1 Schematic illustration of the literature search and the study selection criteria.** CINAH, the cumulative index to nursing and allied health; NTLTD, the net worked digital library of theses and dissertations.

All the interventions involved listening to music. In most studies, sedative, relaxing, stable, and regular rhythmic patterns were recommended or chosen as the music types. Specific musical pieces or songs (the Four Seasons by Vivaldi and Lullaby by Mozart) were mentioned in one study. The use of traditional music, such as Turkish music (Acemasiran mode), Spanish style guitar, Israeli tunes, and Chinese religious music, were mentioned in six studies. Self-selected music, under recommendation, was used in most studies (nine studies; no clear information in two studies).

The time of the music intervention included the latent phase, active phase, and whole stages of labor in studies for participants undergoing normal spontaneous delivery (NSD) and before entering the operation room and during the operation for participants undergoing CS.

For participants undergoing NSD, the intervention duration varied from at least 30–20-min break per 1–2 h during all the labor stages and from 40 min before the surgery to the entire operation duration since entering operation room to the end of surgery for participants undergoing CS.

The VAS-A was used to assess the anxiety status in seven studies, STAI was used in four studies, and Zung's SAS was used in two studies. Physiological indexes, such as SBP, DBP, RR, HR, and $SpO_2$ were recorded in six studies. HRV was tested in one study and cortisol level was used in two other studies. The timing of the outcome assessment varied from right after intervention to 24 h after the intervention in the NSD group and 6 h after the operation in the CS group.

A total of 1,310 participants were enrolled in these studies. Majority of the included studies had a low to moderate potential for bias, as demonstrated by our quality assessment using the Cochrane assessment tool (Table S3).

All the studies compared the intervention and control groups; however, one study incorporated a four-arm design, with two additional groups receiving acupressure intervention as well as acupressure and music combined treatment (Wan & Wen, 2018); one study used a three-arm design, with one additional ear-plug group (Choubsaz et al., 2018).

## Data synthesis and meta-analyses

We focused on the effect of music intervention on the change in the anxiety status. Data pertaining to anxiety evaluated using recognized rating scales including VAS-A, STAI, and SAS and physiological indexes were extracted for further meta-analyses. Studies with different outcome measurements, such as pain scores, depression scores, and satisfaction scores, were excluded. VAS-A was measured in seven studies, STAI in four, SAS in two, and HR and BP in six. However, data from some of the studies (Liu, Chang & Chen, 2010; Gokyildiz Surucu et al., 2018; Kushnir et al., 2012) were not available because the outcomes were measured before delivery, rather than post-delivery. Moreover, data from Choubsaz et al. (2018) was not usable due to uneven pre-test condition. We also excluded Blackburn et al. (2011) due to insufficient data. Finally, nine studies were included for anxiety level meta-analysis (Chang & Chen, 2005; Gokyildiz Surucu et al., 2018; Hepp et al., 2018; Karkal, Kharde & Dhumale, 2017; Li & Dong, 2012; Reza et al., 2007; Simavli et al., 2014a, 2014b; Wan & Wen, 2018) and four studies (Chang & Chen, 2005; Hepp et al., 2018; Lee et al., 2004; Simavli et al., 2014a) for BP and HR meta-analysis. Ultimately, 10 studies with a combined study population of 1,080 participants were included in our meta-analysis.

### Primary outcome

In nine selected studies, the anxiety level of 519 participants in the music intervention group decreased (standardized mean difference (SMD) = −2.40, 95% CI [−3.29 to −1.52], $p < 0.001$; $I^2 = 97.66\%$; Fig. 2) as compared to that in the 511 participants in the placebo group. Although there was publication bias ($t(8) = 4.41$, $p = 0.002$), the results of the meta-analysis did not change (SMD = −4.12, 95% CI [−5.93 to −2.31]) after the trim and fill test (with three potentially missing studies to the left of the mean). The significance remained unchanged after removing any of the studies. Among these nine studies, five were on pregnant women undergoing NSD, and four were on those scheduled for CS. A subgroup analysis, based on the methods of delivery, showed a significant decrease in the anxiety score with music intervention in both, the NSD group (SMD = −4.69,

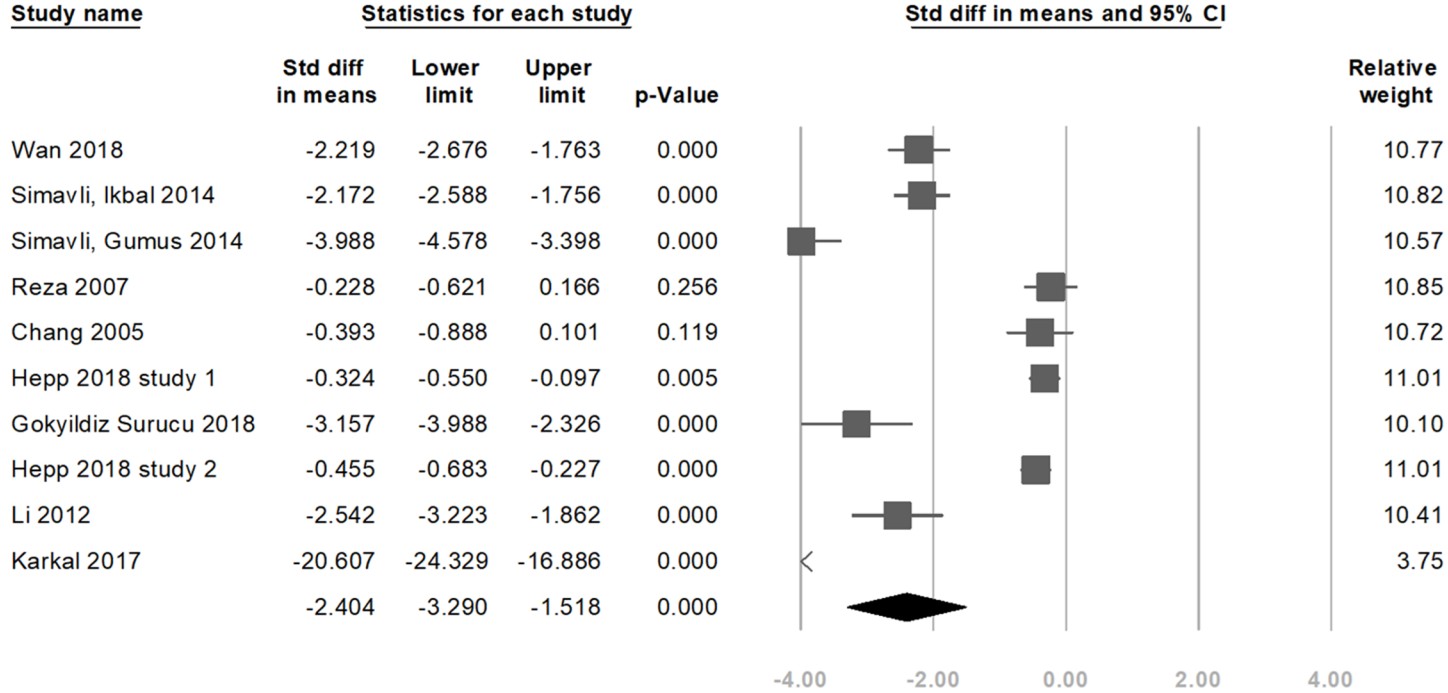

**Figure 2** Forest plot of pooled anxiety scores after the intervention between the music intervention group and the placebo group (Overall meta-analysis).

95% CI [−6.28 to 3.10], $p < 0.001$; $I^2 = 96.63\%$) and CS group (SMD = −0.70, 95% CI [−1.18 to −0.22], $p = 0.004$; $I^2 = 89.69\%$; Fig. 3). Another subgroup analysis by music types showed significant differences if music types were chosen by the participants (SMD = −1.71, 95% CI [−2.59–0.82], $p < 0.001$; $I^2 = 97.48\%$; Fig. 4).

*Secondary outcome*

Combining the results of four studies in relation to physiological indexes, the meta-analysis indicated that those in the intervention group exhibited a significant decrease in the HR (difference in means = −3.04 beats/min, 95% CI [−4.79 to −1.29] beats/min, $p = 0.001$; $I^2 = 0.00\%$; Fig. 5), SBP (difference in means = −3.71 mmHg, 95% CI [−7.07 to −0.35] mmHg, $p = 0.031$; $I^2 = 58.47\%$; Fig. 6), and DBP (difference in means = −3.54 mmHg, 95% CI [−5.27 to −1.81] mmHg, $p < 0.001$; $I^2 = 0.00\%$; Fig. 7).

Subgroup analyses by methods of delivery showed that in the CS group, HR and DBP decreased significantly (HR: difference in means = −2.97 beats/min, 95% CI [−4.80 to −1.15] beats/min, $p = 0.001$; $I^2 = 0.00\%$; Fig. S1A and DBP: difference in means = −3.02 mmHg, 95% CI [−5.49 to −0.54] mmHg, $p = 0.017$; $I^2 = 0.00\%$; Fig. S1C). However, the SBP did not decrease significantly (difference in means = −2.96 mmHg, 95% CI [−7.15–1.23] mmHg, $p = 0.166$; $I^2 = 54.40\%$; Fig. S1B).

Subgroup analyses by music types showed that if the music type was chosen by the participants, HR and DBP significantly decreased (HR: difference in means = −2.85 beats/min,

| Group by Delivery | Study name | Std diff in means | Lower limit | Upper limit | p-Value | Std diff in means and 95% CI | Relative weight |
|---|---|---|---|---|---|---|---|
| a. NSD | Wan 2018 | -2.219 | -2.676 | -1.763 | 0.000 | | 22.73 |
| a. NSD | Simavli, Ikbal 2014 | -2.172 | -2.588 | -1.756 | 0.000 | | 22.81 |
| a. NSD | Simavli, Gumus 2014 | -3.988 | -4.578 | -3.398 | 0.000 | | 22.45 |
| a. NSD | Gokyildiz Surucu 2018 | -3.157 | -3.988 | -2.326 | 0.000 | | 21.79 |
| a. NSD | Karkal 2017 | -20.607 | -24.329 | -16.886 | 0.000 | | 10.22 |
| a. NSD | | -4.689 | -6.280 | -3.098 | 0.000 | | |
| b. CS | Reza 2007 | -0.228 | -0.621 | 0.166 | 0.256 | | 20.36 |
| b. CS | Chang 2005 | -0.393 | -0.888 | 0.101 | 0.119 | | 18.85 |
| b. CS | Hepp 2018 study 1 | -0.324 | -0.550 | -0.097 | 0.005 | | 22.41 |
| b. CS | Hepp 2018 study 2 | -0.455 | -0.683 | -0.227 | 0.000 | | 22.39 |
| b. CS | Li 2012 | -2.542 | -3.223 | -1.862 | 0.000 | | 15.99 |
| b. CS | | -0.701 | -1.181 | -0.222 | 0.004 | | |

Favor intervention          Favor placebo

**Figure 3 Forest plot of pooled anxiety scores after the intervention between the music intervention group and the placebo group (Subgroup analysis by methods of delivery).**

| Group by music type | Study name | Std diff in means | Lower limit | Upper limit | p-Value | Std diff in means and 95% CI | Relative weight |
|---|---|---|---|---|---|---|---|
| a. Patient choice | Wan 2018 | -2.219 | -2.676 | -1.763 | 0.000 | | 14.31 |
| a. Patient choice | Simavli, Ikbal 2014 | -2.172 | -2.588 | -1.756 | 0.000 | | 14.40 |
| a. Patient choice | Simavli, Gumus 2014 | -3.988 | -4.578 | -3.398 | 0.000 | | 13.95 |
| a. Patient choice | Chang 2005 | -0.393 | -0.888 | 0.101 | 0.119 | | 14.21 |
| a. Patient choice | Hepp 2018 study 1 | -0.324 | -0.550 | -0.097 | 0.005 | | 14.73 |
| a. Patient choice | Hepp 2018 study 2 | -0.455 | -0.683 | -0.227 | 0.000 | | 14.73 |
| a. Patient choice | Li 2012 | -2.542 | -3.223 | -1.862 | 0.000 | | 13.67 |
| a. Patient choice | | -1.705 | -2.589 | -0.820 | 0.000 | | |
| b. Not patient choice | Reza 2007 | -0.228 | -0.621 | 0.166 | 0.256 | | 100.00 |
| b. Not patient choice | | -0.228 | -0.621 | 0.166 | 0.256 | | |

Favor intervention          Favor placebo

**Figure 4 Forest plot of pooled anxiety scores after the intervention between the music intervention group and the placebo group (Subgroup analysis by music types).**

| Study name | Statistics for each study | | | | Difference in means and 95% CI | Relative weight |
|---|---|---|---|---|---|---|
| | Difference in means | Lower limit | Upper limit | p-Value | | |
| Simvali, Gumus 2014 | -2.880 | -5.306 | -0.454 | 0.020 | | 52.02 |
| Chang 2005 | -3.810 | -9.838 | 2.218 | 0.215 | | 8.43 |
| Lee 2004 | -5.000 | -10.851 | 0.851 | 0.094 | | 8.95 |
| Hepp 2018 | -2.540 | -5.703 | 0.623 | 0.116 | | 30.61 |
| | -3.044 | -4.794 | -1.294 | 0.001 | | |

-8.00   -4.00   0.00   4.00   8.00

Favor intervention      Favor placebo

**Figure 5 Forest plot of heart rate after the intervention between the music intervention group and the placebo group.**

| Study name | Statistics for each study | | | | Difference in means and 95% CI | Relative weight |
|---|---|---|---|---|---|---|
| | Difference in means | Lower limit | Upper limit | p-Value | | |
| Simvali, Gumus 2014 | -5.540 | -9.020 | -2.060 | 0.002 | | 30.43 |
| Chang 2005 | -4.660 | -12.294 | 2.974 | 0.232 | | 13.57 |
| Lee 2004 | -6.000 | -11.072 | -0.928 | 0.020 | | 22.27 |
| Hepp 2018 | -0.160 | -3.070 | 2.750 | 0.914 | | 33.74 |
| | -3.708 | -7.070 | -0.346 | 0.031 | | |

-8.00   -4.00   0.00   4.00   8.00

Favor intervention      Favor placebo

**Figure 6 Forest plot of systolic blood pressure after the intervention between the music intervention group and the placebo group.**

95% CI [−4.69 to −1.02] beats/min, $p = 0.002$; $I^2 = 00.00\%$; Fig. S1D and DBP: difference in means = −3.48 mmHg, 95% CI [−5.32 to −1.65] mmHg, $p < 0.001$; $I^2 = 00.00\%$; Fig. S1F). However, the SBP did not decrease significantly (difference in means = −3.09 mmHg, 95% CI [−7.17–0.99] mmHg, $p = 0.137$; $I^2 = 65.03\%$; Fig. S1E).
| Study name | Statistics for each study | | | | Difference in means and 95% CI | |
| --- | --- | --- | --- | --- | --- | --- |
| | Difference in means | Lower limit | Upper limit | p-Value | | Relative weight |
| Simvali, Gumus 2014 | -4.040 | -6.452 | -1.628 | 0.001 | | 51.33 |
| Chang 2005 | -3.430 | -9.710 | 2.850 | 0.284 | | 7.57 |
| Lee 2004 | -4.000 | -9.149 | 1.149 | 0.128 | | 11.26 |
| Hepp 2018 | -2.540 | -5.703 | 0.623 | 0.116 | | 29.84 |
| | -3.542 | -5.270 | -1.814 | 0.000 | | |

-8.00    -4.00    0.00    4.00    8.00

Favor intervention        Favor placebo

**Figure 7** Forest plot of diastolic blood pressure after the intervention between the music intervention group and the placebo group.

There was no publication bias ($t(2) = 1.98$, 0.94, and 0.20; $p = 0.19$, 0.44, and 0.86 for HR, SBP, and DBP, respectively). However, after removing data from either of the studies, except one study (*Hepp et al., 2018*) that included the largest population, the change in SBP was not significantly different between the music intervention and placebo groups (*Simavli et al., 2014a*: difference in means = −2.96 mmHg, 95% CI [−7.15–1.23] mmHg, $p = 0.166$; *Chang & Chen, 2005*: difference in means = −3.62 mmHg, 95% CI [−7.66–0.41] mmHg, $p = 0.078$; and *Lee et al., 2004*: difference in means = −3.09 mmHg, 95% CI [−7.17–0.99] mmHg, $p = 0.137$). Moreover, the funnel plots were also assessed (Fig. S2).

We performed a meta-regression analysis using age as the moderator in the single meta-regression to examine the heterogeneity of the present analysis. The result showed that the effect of music intervention on the anxiety level was not significantly confounded by age (slope = 0.029, $p = 0.723$; Fig. 8).

## DISCUSSION

Our systematic review and meta-analysis support the beneficial effects of music intervention on anxiety during labor, in subjective and objective dimensions. We found that music intervention decreased anxiety score, HR, SBP, and DBP after the intervention (−2.40 SMD, 3.04 beats/min, 3.71 mmHg, and 3.54 mmHg, respectively).

The present meta-analysis comprehensively investigated the efficacy of music intervention on anxiety during labor. The majority of the recent meta-analyses focuses on pain (*Smith et al., 2018*) or music intervention during pregnancy (*Corbijn Van Willenswaard et al., 2017*). Other than pain, high anxiety levels also cause multiple negative effects, such as elevated BP, increased cortisol levels, elevated HR, slower wound healing, reduced immune response, increased infection risk (*Scott, 2004*), and enhanced anesthesia

## Regression of standardized mean difference on Age

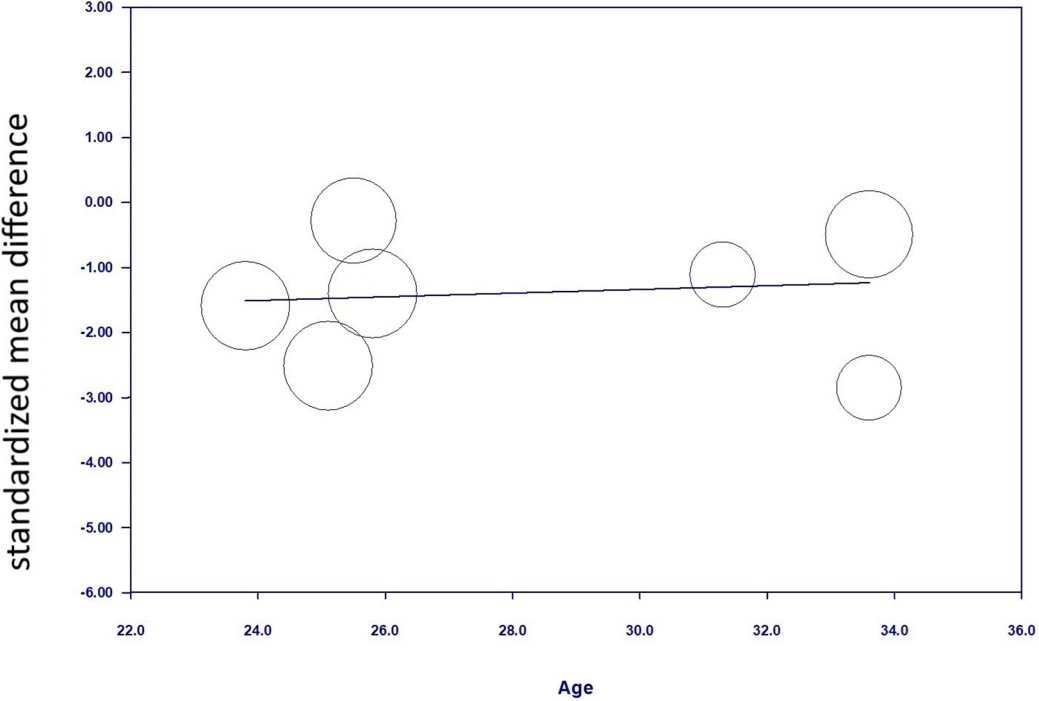

**Figure 8 Regression of anxiety scores on Age.**

induction difficulty (*Özalp et al., 2003*). Furthermore, anxiety of being awake during operations has been shown to be the main reason for choosing general anesthesia (*Shevde & Panagopoulos, 1991*). Higher risk and more adverse effects on the mother and child during sedation have been well known; therefore, non-pharmaceutical intervention for anxiety during labor is important. Our survey focused on studies designed to evaluate the effect of music intervention during the labor process on anxiety levels evaluated after childbirth.

Our results are consistent with the previous studies related to the effects of music intervention on lower anxiety in participants during labor (*Chuang et al., 2018*). Nevertheless, the analysis of anxiety level by *Chuang et al. (2018)* only involved one study that evaluated the anxiety level using VAS-A and one study that used the Zung SAS on primiparous women expected to undergo NSD. In our analysis, we pooled six studies that evaluated the anxiety level with VAS-A, two studies with SAS, two studies with STAI, and included pregnant women prepared to deliver via either NSD or CS. We used strict inclusion criteria (only RCTs in the meta-analysis) and recent studies rather than the old ones. *Chuang et al. (2018)* did not execute subgroup analysis as per the method of delivery. In the current study, music intervention successfully lowered the anxiety score in women who delivered via NSD as well as those who underwent CS (Fig. 3).

*Reza et al. (2007)* reported no significant decrease in the VAS-A score in CS participants; low level of preoperative anxiety in the study was considered a possible explanation (*Nilsson et al., 2001*). Our meta-analysis results revealed a significant decrease in the anxiety score, HR, and DBP after the music intervention in the CS group, but to lesser

extent than in NSD group (SMD of anxiety score in CS group = −0.70, SMD = −4.69 in NSD group; Fig. 3), which is consistent with a previous meta-analysis, according to which, music during planned CS may improve the pulse rate and birth satisfaction score with a small magnitude (*Laopaiboon et al., 2009*). Another study reported that participants undergoing surgery with local anesthesia who listened to music during surgery had significantly lower HR, anxiety, and BP (*Mok & Wong, 2003*).

Another possible reason for the ineffectiveness of music intervention in the study by *Reza et al. (2007)* may be related to the type of music. Participants did not have the opportunity to preoperatively choose the type of music and a culturally unfamiliar type of light music was used in their study. Perioperative music intervention changed the neurohormonal and immune stress response, especially if the participants selected the type of music by themselves in a study on participants undergoing day surgery (*Leardi et al., 2007*). *Aldridge (1994)* declared that the effects of music were influenced by how much the subjects appreciated the type of music. Women were suggested to develop individual preferences for the use of music and equipment (*Gentz, 2001*). In our meta-analysis, significant decreases in the anxiety score, HR, and DBP were found after the music intervention as compared to after placebo treatment, if the music type was chosen by the participants (Fig. 4), which is consistent with the findings of the previous studies.

Several studies have evaluated the effect of music on the cardiovascular system (*Koelsch & Jäncke, 2015*). The levels of BP and HR are reflected in terms of stress and anxiety other than subjective parameters (*Koelsch & Jäncke, 2015*). The music intervention group showed significantly lower HR and BP levels in our meta-analysis with low heterogeneity. The outcomes support the positive findings of subjective parameters.

The mechanism of music to affect physiological indicators of anxiety is based on the psychophysiological theory (*O'Callaghan, Sexton & Wheeler, 2007*). Music may relief anxiety by stimulating pleasure, distracting concentration, and providing a bridge for meditation (*Browning, 2000*; *Phumdoung & Good, 2003*). Music can activate the release of endorphins, which are the body's pain killers, to lower the unpleasant feelings and emotions and also lower the sympathetic nervous system activity, HR, BP, respiratory rate, oxygen consumption, metabolic rate, skeletal muscle tension, sweat gland activity, blood epinephrine level, plasma prednisone level, number of natural killer cells, neurohormonal stress, and immune stress (*Arslan, Özer & Özyurt, 2008*). This may explain the changes in the physiological indicators observed in the intervention group in our study.

The effect of music intervention on the anxiety score was not significantly confounded by age when age was used as a moderator in the single meta-regression. *Rubertsson et al. (2014)* reported that women <25 years of age had a higher risk of anxiety symptoms during early pregnancy. Other risk factors of anxiety included the use of different languages, lower education level, unemployment, nicotine use before pregnancy, and a self-reported psychiatric history of either depression or anxiety before the current pregnancy. The mean participant age in our study ranged from 23.8 to 33.6 years. A mean participant age <25 years was only found in one study.

The strengths of our study included a relatively larger total study population than that in other reviews on similar topics and the lack of language restriction. Considering our

results, there are great indications for clinical practice. Music is inexpensive, effective, safe, and easy to be used in daily clinical practice. Therefore, music interventions can be offered as a routine practice to women during labor.

There are certain limitations to this study. First, some subgroup analyses were not performed due to the lack of studies, such as those regarding music intervention time, and the time of outcome measurement. Second, the number of trials included was limited, making it difficult to execute more meaningful meta-regression analyses to examine the impact of variables that may affect the heterogeneity of the results. Third, the sample size of some included studies was small and did not provide details on the randomization processes. Fourth, the time of outcome measurement and music types in each study varied, thus potentially limiting the usability of some results. We attempted to select the result with a similar time of outcome measurement from each study; however, there remained high heterogeneity in the meta-analysis of the pooled anxiety scores after the intervention between the music intervention and the placebo groups, included in the subgroup analyses. The anxiety level changed largely during the active phase; therefore, when evaluating the treatment effects, the time of the assessment was important. Finally, the variation between these studies in terms of participant characteristics, intervention design, and time of outcome measurements should be considered. Further well-designed studies are needed to clarify the influences of music types, time of music intervention, and time of outcome measurement on the effect of music intervention.

## CONCLUSIONS

Thus music interventions during labor significantly reduce the anxiety scores and physiological indexes related to anxiety (HR, SBP, and DBP). Music interventions may be effective in reducing the anxiety levels during labor. Application in clinical routine may be advisable. Additional large RCTs focusing on the music types, time of music intervention, and time of outcome measurement are required to validate these findings.

## ACKNOWLEDGEMENTS

The authors would like to thank Enago, the editing brand of Crimson Interactive Pvt., Ltd. for the English language review.

### Funding

The authors received no funding for this work.

### Competing Interests

The authors declare that they have no competing interests.

### Author Contributions

- Hsin-Hui Lin analyzed the data, contributed reagents/materials/analysis tools, prepared figures and/or tables, approved the final draft.
- Yu-Chen Chang prepared figures and/or tables.

- Hsiao-Hui Chou prepared figures and/or tables.
- Chih-Po Chang prepared figures and/or tables.
- Ming-Yuan Huang contributed reagents/materials/analysis tools.
- Shu-Jung Liu contributed reagents/materials/analysis tools.
- Chin-Han Tsai conceived and designed the experiments.
- Wei-Te Lei analyzed the data, contributed reagents/materials/analysis tools, prepared figures and/or tables, authored or reviewed drafts of the paper, approved the final draft.
- Tzu-Lin Yeh conceived and designed the experiments, contributed reagents/materials/analysis tools, authored or reviewed drafts of the paper, approved the final draft.

### Data Availability
Raw data are available in the Supplemental Files.

### Supplemental Information
Supplemental information for this article can be found online at http://dx.doi.org/10.7717/peerj.6945#supplemental-information.

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
