# Peer review of "Effect of music interventions on anxiety during labor: a systematic review and meta-analysis of randomized controlled trials"

_PeerJ, doi:10.7717/peerj.6945_

## Round 0.1 · original submission · Major Revisions

This is an interesting meta-analytic review. There are however many issues identified by our reviewers that need to be addressed for this can be more seriously considered for publication. Please be sure to revise the paper carefully.

Reviewer 1 ·

Basic reporting

In general, this manuscript is well written. But I found two grammatical errors in line 201 and line 305. “Total 1,310 participants” and “Aldridge et al declared that” should change it to “A total of 1,310 participants” and “Aldridge et al. declared that”, respectively.

Experimental design

The greatest problem in this paper is that the time frame of data retrieval is from April 25, 2018 to the end of December, 2018. For a meta-analysis, this is not a comprehensive search for relevant literature.

Validity of the findings

No comment

Additional comments

This is a meaningful, interesting study in which the authors confirmed that music interventions may decrease the anxiety levels measured using the VAS-A and physiological indexes related to anxiety (HR, SBP, and DBP). However, some issues should be solved. The specific points need to be carefully corrected as follows:

Abstract
-- Authors need to report specific the name of databases and time frame in this part.

Introduction
-- It is necessary to introduce some basic information about music intervention in this part.

Method
-- All abbreviated name should be corrected. For example, “H.-H.L.” change it to “HHL”.
-- I suggest that the authors should write the inclusion/exclusion criteria based on the “PICOs” approach.
-- Line 113-114. “without deafness (unless corrected with a hearing aid) or severe psychiatric disorder” This is an exclusion criterion rather than an inclusion criterion.
-- Please cite the references for related heterogeneity criteria.

Result
--Authors should describe basic feature of the included studies in brief. The best way is to perfect table 1 rather than a detailed and lengthy description.
-- The references of the included studies should only be listed once.
-- For subgroup analyses, results from comparisons across groups can provide more useful information.

·

Basic reporting

no comment

Experimental design

I did not have access to the additional tables and figures, which was lacking in the review process. If they have not been placed in the submission process, it is essential that they be sent.
The term music therapy is used several times and appears in the titles of some figures, but the authors do not make clear the difference between music therapy (performed by a professional Music Therapist) and musical interventions. This differentiation is of fundamental importance considering that the term Music therapy appears at different points in time. Questioned? Was a study differentiated with music therapy and music in medicine? Was Music Therapy considered as a musical intervention? How many and which studies were with Music Therapy? I consider it important to clarify this aspect as well as information on how the data were analyzed considering that there is a difference between these practices.

Validity of the findings

The validity of the data should be better analyzed after clarification on experimental design

Additional comments

The article was well written and there is only one aspect of doubt pointed out in the design of the study. This area lacks studies of this nature, which allows this to be an essential publication.
I did not have access to the additional tables and figures, which was lacking in the review process. If they have not been placed in the submission process, it is essential that they be sent.

Reviewer 3 ·

Basic reporting

no comment

Experimental design

no comment

Validity of the findings

no comment

·

Basic reporting

English language editing is necessary - especially in chapters "Introduction" (e.g. l. 62; l. 72-73, l. 79) and "Discussion"
References: The references given in the introduction section are not quite fitting. Nearly all references in the introduction are questionable, because the topics do not fit or the quoted studies are irrelevant for the current context. A number of studies is out of date (published before 2000 but not of outstanding scientific impact). The current references refer to completely different fields, such as ICU, chronic pain or cancer.
I would strongly recommend to stick to the current topic (labor pain and anxiety), you might back up the introduction with some general information on music + analgesia and music + anxiety (there exists a number of high quality reviews and meta-analysis on this topic).
In the discussion section, a rather comprehensive overview, that fits to the current topic is included - Maybe you could restructure the paper and use several paragraphs from the discussion rather in the introduction section!!

Background: Please, give a precise definition of "music therapy". The term is used in a very unspecific manner, therefore it is very important to make clear, what you mean by "music therapy". There exist good definitions by the American Music Therapy Association (AMTA) https://www.musictherapy.org/
or the World Federation of Music Therapy (WFMT) https://www.wfmt.info/

Article structure, figures, tables:
- The structure of the article is conform to an acceptable format of ‘standard sections’
- All appropriate raw data have been made available in accordance with the Data Sharing policy.
- Figures are relevant to the content of the article; It would be helpful to add a label to the figures in the supplement, in order to make sure, the figures are comprehensible in a stand alone version.

Experimental design

Search strategy:
- did you search for a specific period? Please state the space of time, studies would be eligible.
Data extraction:
- why did you choose the outcome variables you chose? This should be explained in more detail.
- what do you mean by "visual analoge scale"? Please give more details on the operationalization (e.g. " It is a scale that is comprised of a horizontal or vertical line and anchored at both ends by words indicative of extremes of magnitude" - if the studied did not measure their data according to this definition but askes participants to rate their mood on a scale 0-100% or allocate it to a number 0-10, this would not be a VAS but rather a NRS (numeric rating scale) - please be precise here!
Did all scales refer to a range of 0-10? Did you convert any different measures to this scale? Please be more precise on this topic!
- the what about the psychometric more sophisticated questionnaires? Why did you not follow up the data of these questionnaires? A strengh of a meta-analysis can be to pool outcomes from different sources!
- l. 127: "The secondary outcomes were physical signs, such as BP and HR." ==> please, be precise in naming the secondary Outcomes!
- Control Group --> inconsistent Information. L. 114: "2) inclusion of a control Group" vs. L. 117/118: "4) studies with an effective intervention as control arm rather than a Placebo" - please be precise here. Why did you exclude studies with an effective intervention as control arm rather than a Placebo?
- l. 139-141: "A p value < 0.05 was considered significant. Statistical heterogeneity was assessed using I2 and Cochran’s Q tests. A p value < 0.10 for the χ2 test of the Q statistic or an I2 > 50% was considered indicative of statistically significant heterogeneity." ==> why did you use different alpha-Levels?= please explain or adujst!

Validity of the findings

- It is unclear why the results restrict to 14 studies overall and to 7 (or 6?? l. 217) studies in the meta-Analysis. Please be again more precise in describing you Methods.
- Did you weigh the studies? If yes, please explain your algorithm!
- l. 172/1733: "Self-selected music under recommendation was used in most studies (at least 9 studies" --> please give exact numbers!
- l.155-159: why did you stress the origin of the studies? Is is rather unusual to have so many Asian studies??

l. 213-217: "Ultimately, 7 studies ... were included in our meta-analysis... Primary Outcome: in 6 selected studies" --> what happended to study #7?

l. 222: "t value = 1.18, df = 4, p = 0.304" --> unusual report, rather 't-test: t(4) = 1.18, p = .304'

l. 230-231: "difference in means −= −0.28" unclear ??


- the trials included in the review are very, very heterogeneous. It therefore is doubtful to conduct a meta-Analysis on the sparse data.
I would suggest to emphasize the narrative review rather than extracting data of feeble informative value.

- l. 260-263: why did you emphasize age? Other variables might have been
number of birth (firstborn, later-born)

ethnic/cultural Background (there exists evedence concerning differences in pain and anxiety perception in different cultures)
Family Status (married/single etc., big Family Background,single-child-family…)

Additional comments

- the paper Reports interesting Facts concerning the use of prerecoded Music on anxiety Levels of Women during Labor.

- make sure to report new findings and to distinguish your results from previous, often quite similar papers.

Major concerns are
1) the choice of papers for the review/meta-analysis and the chosen outcome criteria --> please be more precise in the methods section. Currently you put quite an effort into the attempt to avoid any bias but you do not manage to outline your search and selection process.
2) the background section is completely inconclusive, since nearly all papers quoted do not refer to the topic "labor+anxiety and music" --> a complete revision of the introduction section is inevitable!
3) the results of the meta-analysis are very flabby, due to the heterogenity and the very small number of the raw Studie --> rather emphasize the narrative review instead of statistical fireworks without clinical relevance

---

## Round 0.2 · Minor Revisions

Thank you very much for the efforts in revising the paper. The three reviewers are satisfied with this version. However, one reviewer still has a minor comment for it. Please kindly correct it. Before your re-submission, please carefully check your manuscript again according to PeerJ's guideline for authors.

Reviewer 1 ·

Basic reporting

No

Experimental design

NO

Validity of the findings

NO

Additional comments

I am satisfied with the changes the authors made.

·

Basic reporting

The authors met the requests of the reviewers; therefore I consider the article suitable for publication. They updated some references, and made the table title corrections.

Experimental design

The authors responded to the requests of the reviewers.

Validity of the findings

The authors responded to the requests of the reviewers.

·

Basic reporting

Clear and unambiguous, professional English used throughout.


Literature references, sufficient field background/context provided.


Professional article structure, figures, tables. Raw data shared.


Figures should be relevant to the content of the article, of sufficient resolution, and appropriately described and labeled.


Self-contained with relevant results to hypotheses.

Experimental design

Design improved, no further comments

Validity of the findings

Validity of findings greatly improved

Additional comments

Many thanks for your effort in implementing the suggestions. The article improved greatly and in my opinion does not further re-editing.
One minor comment: in the section "data extraction" you mention secondary outcomes, i.e. physical signs. Please spell out the abbreviations since they appear for the first time in the running text (l. 145 in the tracked changes document)

---

## Round 0.3 · accepted · Accept

Thanks for your revision.

#